# Minerals in Pregnancy and Their Impact on Child Growth and Development

**DOI:** 10.3390/molecules25235630

**Published:** 2020-11-30

**Authors:** Patricia Miranda Farias, Gabriela Marcelino, Lidiani Figueiredo Santana, Eliane Borges de Almeida, Rita de Cássia Avellaneda Guimarães, Arnildo Pott, Priscila Aiko Hiane, Karine de Cássia Freitas

**Affiliations:** 1Graduate Program in Health and Development in the Central-West Region of Brazil, Medical School, Federal University of Mato Grosso do Sul, Campo Grande 79070-900, Mato Grosso do Sul, Brazil; patmiranda_nut@hotmail.com (P.M.F.); gabi19ac@gmail.com (G.M.); lidi_lfs@hotmail.com (L.F.S.); rita.guimaraes@ufms.br (R.d.C.A.G.); priscila.hiane@ufms.br (P.A.H.); 2Biologist, Hematology Laboratory, State Secretariat of Health of Mato Grosso do Sul, Campo Grande 79084-180, Mato Grosso do Sul, Brazil; elianeba.ms@gmail.com; 3Graduate Program in Biotechnology and Biodiversity in the Central-West Region of Brazil, Federal University of Mato Grosso do Sul, Campo Grande 79070-900, Mato Grosso do Sul, Brazil; arnildo.pott@gmail.com

**Keywords:** children, minerals, iodine, selenium, iron, zinc, calcium, magnesium

## Abstract

During pregnancy, women undergo metabolic and physiological changes, and their needs are higher, to maintain growth and development of the fetus. If the nutritional status of the expectant mother is not satisfactory, some maternal and neonatal complications can occur. In the second and third trimester of pregnancy, there is a reserve of nutrients in the fetus that can be utilized after birth; thereby, children present an accelerated growth in the first years of life, which is a proven response to the available nutrition pattern. However, if such a pattern is insufficient, there will be deficits during development, including brain function. Therefore, despite many recent published works about gestational nutrition, uncertainties still remain on the mechanisms of absorption, distribution, and excretion of micronutrients. Further elucidation is needed to better understand the impacts caused either by deficiency or excess of some micronutrients. Thus, to illustrate the contributions of minerals during prenatal development and in children, iodine, selenium, iron, zinc, calcium, and magnesium were selected. Our study sought to review the consequences related to gestational deficiency of the referred minerals and their impact on growth and development in children born from mothers with such deficiencies

## 1. Introduction

Pregnancy, from conception to birth, is a period when women undergo metabolic and physiological changes, such as weight gain, of which 60% can be attributed to maternal alterations and 40% refers to the placenta and the fetus. During pregnancy, the nutritional needs are higher; thus, adequate nutrition is essential to maintain fetal growth and development. However, pregnancy under inappropriate nutritional conditions can cause maternal and neonatal complications [1,2]. Among these complications is gestational diabetes mellitus, where there is a change in the metabolism of macronutrients, insulin resistance resulting in maternal weight gain, and risk of delivery by cesarean section and babies with macrosomy [3].

A gestational diet poor in nutrients can lead to maternal and fetal malnourishment, which can be observed from birth until infancy, and can result in premature births with gestational ages below 37 weeks, low birth weight (<2500 g), and babies that are small for gestational age (SGA) [4,5]. During infancy, this can lead to stunted growth, neurologic and cognitive delay, as well as alterations in cardiometabolic, pulmonary, and immune function, in addition to complications for the mother such as mortality and pre-eclampsia [4,5].

Absorption, distribution, and excretion of micronutrients during pregnancy are in higher demand due to increased maternal blood volume and metabolism, kidney function, and circulating hormones [6]. The fetus since conception evolves in its growth according to the maternal nutrition through the placenta, and if the diet is inadequate, then supply to the fetus will be limited and can lead to a chemical competition between mother and baby [2]. To increase the long-term maternal reproductive potential, the body sometimes understands that it is necessary to sacrifice the current fetus, which can occur even in supplemented mothers [2].

Food ingestion is observed during pregnancy to verify reduction or excess of macro- and micronutrients, since they can be associated with complications in pregnancy and neonatal health [1,7]. Evidence suggests that deficiency of fetal micronutrients persists for generations, with possible intergenerational consequences [1,8]. Disease development in adult life is, in fact, the result of an inadequate distribution of nutrients during fetal life, gestation, and early infancy. Thus, children will show stunted development of muscles, nephrons, and bones, and if the feeding habit of the child contributes to weight gain, they can also, as a consequence, develop insulin resistance and chronic diseases such as type 2 diabetes and metabolic syndrome [2].

Pregnant women in low-income countries generally have nutrient-poor diets [9]. Usually, caloric undernourishment in these expectant mothers occurs through malnutrition, a lack of adequate food diversity, and consumption of many processed foods and not enough fruits, vegetables, and greens [2]. Besides social determinants, lack of physical activity, not taking multivitamins, ingestion of coffee, alcohol, and exposure to tobacco also contribute to unsatisfactory results in pregnancy [6].

Besides expectant mothers, micronutrient deficiencies also affect children under five years, who are at higher risk mainly in low- and medium-income countries, as they lack access to adequate foods rich in vitamins and minerals, health services, and hygiene care, and they live in unhealthy environments that promote increased susceptibility to infections, inflammation, and disease [8]. Micronutrients are essential to the human organism and may vary in quantities necessary for each life phase, physiological function, and for health maintenance [5,8]. Micronutrient deficiency in industrialized and developing countries affects over 2 billion people in all age groups and is responsible for 10% of child mortality [10].

Among the deficient micronutrients, iron, iodine, folate, zinc, and vitamin A are the most widespread since they commonly contribute to growth, and in their absence perinatal complications, intellectual disorders, and increased risk of morbimortality are observed [8].

Despite the recent number of published studies on gestational nutrition, many uncertainties still remain regarding the mechanisms of micronutrient absorption, distribution, and excretion. Further elucidation is needed to better understand the impacts caused either by deficiency or excess of some micronutrients. This literature review is timely because we still see many newborns who are born with low weight, have problems in neuropsychomotor development, and have disorders of hyperactivity. It is necessary to further study these processes and show the importance of an adequate intake of micronutrients for proper development of an individual and the implications for health. As we will see throughout the text, improper development can be irreversible.

To illustrate the contributions of minerals to prenatal and child development, iodine, selenium, iron, zinc, calcium, and magnesium were selected. Our study aimed to review the consequences of gestational deficiency of these minerals and their impact on child growth and development. The searches were carried out with Medline (PubMed), Lilacs, and Scie Direct databases. At the end of the search in each database, articles related to mesh terms in English—minerals, pregnancy, growth and development, and child—were included. Duplicate references and those that did not include the researched minerals were excluded.

## 2. Micronutrients

Vitamins and minerals, also named micronutrients, are essential to many enzymes, in the form of cofactors and coenzymes that promote maintenance, formation, and homeostasis of body tissues, and to perform metabolic activities, such as cell signaling, motility, proliferation, differentiation, and apoptosis. Generally, human needs are below 100 mg/day, such as with essential vitamins in microgram (µm) or milligram (mg) quantities, whereas macronutrients are needed in higher quantities (g/day) [5,11]. Minerals (macroelements, microelements, or oligo-elements and traces), in addition to some vitamins, cannot be synthesized by the human organism; thus, they must be acquired through foods or, as needed, by supplementation. The major minerals of the body are calcium, sodium, magnesium, phosphorous, and potassium, and they have specific functions in the metabolism of zinc, manganese, molybdenum, iodine, selenium, sulfur, iron, chlorine, cobalt, and copper, also known as oligo-elements [11].

Deficiency or excess of micronutrients during pregnancy can bring irreversible consequences to the newborn and child. If there is severe deficiency in at least one of the nutrients, depending on the length of deprivation, the child may have a decreased antioxidant defense, immune response, redox signaling, wound cicatrization, and expression of regulator genes involved in the development of diseases [11,12]. In addition, child neurodevelopment is also jeopardized, as observed with iodine and iron deficiencies [12].

In the second and third trimester of pregnancy, there is a reserve of nutrients in the fetus that can be utilized after birth; therefore, children present accelerated growth in the first two years of life (the child’s first 1000 days, being defined from conception until the second year of age) [13]. This phase is essential for development and growth and notably is a result of the offered nutritional standard; however, if such standards are insufficient, the child will have developmental deficits, including brain function [13].

Micronutrients, being metabolized at adequate levels, from a maternal reserve allow the fetus to grow and develop healthily [5,14]. The essential oligo-elements such as copper, manganese, selenium, and zinc are relevant for the maintenance of cell proliferation in the formation of the embryo [5]. Copper, zinc, selenium, and iron are vital minerals to support reproduction and prevent poor fetal development and complications at pregnancy [14].

Most research about micronutrient deficiency during pregnancy is on folate and vitamin D [14]. Deficiencies of iron, iodine, and zinc are often overlooked, although they have great impacts on public health, and calcium, fluor, selenium, and magnesium have the highest health costs [11]. We will analyze some minerals to elucidate their biochemical mechanisms in their deficiency during pregnancy and the consequences for both mothers and children.

### 2.1. General Characteristics

#### 2.1.1. Iodine

This mineral is a relevant component of the thyroid hormones tri-iodine-thyronine (T3) and thyroxine (T4), which regulate different biological processes, acting especially on the thyroid gland and the immune system [15,16]. The content of this mineral is variable in foods, being mainly found in marine algae and fish. Thus, populations that include them in their diet can easily reach daily requirements [17]. The recommendations of iodine for adequate production of T4 are variable according to sex and age group [16].

Iodine deficiency is found in all age groups, and lack of iodine leads to hypothyroidism and other disorders, classified as iodine deficiency disorders (IDDs) [17,18]. Iodine supplementation can be recommended to fulfil individual needs, especially for expectant mothers, to avoid possible impacts that its deficiency can have on the neural development of the fetus [19].

Recent data published by UNICEF [18] show that in 2018, 88% of the world population consumed iodized salt: the Far East and Pacific countries presented the highest consumption (92%), followed by South Asia (89%).

Furthermore, iodine deficiency can lead to oxidative stress, causing disturbances in trophoblastic cell function and the placental vascular net. This mineral is also responsible for the redox balance since it has the capacity to compete with free radicals or induce the action of enzymes with antioxidant activity [20].

Iodine also presents a direct relation with another mineral, selenium. When iodine is in excess in the organism, a higher concentration of selenium is demanded from the liver and kidneys. This occurs since iodine diminishes the captivity of glutathione peroxidase (GSH-Px), an antioxidant enzyme dependent on selenium. Thus, the higher the quantity of iodine, the higher the need for selenium in the organism as a mechanism to impede oxidative stress caused by iodine [21]. Furthermore, selenium deficiency affects the thyroxine iodine deiodinases (DIO), another selenoprotein that is responsible for the conversion of thyroid hormones [22,23,24]. DIO1 and DIO2 are located in various tissues, whereas DIO3 can only be found in the placenta [24]. Selenium deficiency can still result in accumulation of peroxides in the thyroid glands, and this leads to the destruction of cells and, thus, possible fibrosis and hypothyroidism [25].

#### 2.1.2. Selenium

Selenium is a mineral that has antioxidant activity. It eliminates free radicals as well as inhibits the production of new ones in the organism since it is an essential element for GSH-Px, which is an enzyme with antioxidant activity [21]. Other functions include its action in the immune system, maintenance of natural defenses, modulation of growth and development, and prevention of heart diseases and cancer [16,26].

Selenium is found in foods such as grains, vegetables, meats, and oil seeds like Brazil nuts and other nuts, with variable concentrations mainly due to environmental conditions. In general, the available selenium in foods is easily absorbed since it has a high bioavailability [16]. A recent review pointed out that the recommended daily doses of selenium vary according to age group as well as the analyzed country, since this mineral occurs in different concentrations according to the soil. The values found for recommendations vary between 45 and 50 µg/day for adults 19 to 50 years, 60 µg/day for women, 70 µg/day for adults, and 75 µg/day for men and nursing mothers [27]. FAO (2001) cites that selenium values below 0.9 µmol/L are responsible for alterations in the balance of thyroid hormones, affecting primarily children and the elderly over 65 years. Evidenced by its high concentrations in the thyroid gland, selenium is required for proper biologic functioning [22].

Selenium deficiency has been more associated with reduced immune system activity (innate and adaptative), precisely the activity of selenoproteins in the oxidative system that eliminate the free oxygen radicals formed in the production of thyroid hormones [23,28]. Selenium deficiency is also related to the development of cretinism since it presents a direct relation with iodine in the organism during hormone conversion [16,23,26,28].

There are 25 types of selenoproteins in humans, one more than found in rodents. Most are expressed in the thyroid hormones and act on the oxidative system, being associated with prevention or development of cancer [22,28,29]. These selenoproteins regulate the expression of enzymes responsible for the conversion of T3 into T4 [22].

Selenium, when at adequate concentrations, is responsible for improving the proliferation of substances that are activated by T cells, as well as to increase the activity of NK cells. It is worth to point out that when selenium deficiency occurs, the literature describes a reduction in T cells that will lead to increased production of oxidant substances [23].

#### 2.1.3. Iron

Iron is an essential mineral for specific activities in the organism, including oxygen transport and production of erythrocytes, it acts as a cofactor in the transport of enzymes (mainly those active in the metabolism of lipids), and it plays a crucial role in the maintenance of the immune system [30,31,32].

Iron deficiency occurs when there is unbalanced demand for iron in the organism in quantities insufficient for homeostasis [33]. This is a worldwide public health problem affecting mainly children in the first infancy, nursing mothers, and women of reproductive age [34,35,36,37].

In this scenario, anemia is the prevalent disorder associated with iron deficiency, and for many years it has been considered one of the most relevant diseases worldwide caused by nutritional deficiencies [35,38]. It is estimated that 50% of pregnant women, 42% of children (under five years), and 33% of women of reproductive age are affected by the disease. Continents such as Africa, Asia, South America, and even Eastern Europe are places where anemia is very present, especially in women of reproductive age [32,36,37]. In general, anemia is linked to inadequate ingestion of iron; however, other factors can be involved such as deficiency of folate and vitamin B12, problems of low absorption, inefficient iron transport, parasite infections, and even other diseases, such as HIV, for example [35,39].

The disease is defined as a low concentration of hemoglobin in the blood, affecting oxygen transport in the organism [37], and the installation of anemia can indeed be considered as the final stage of iron deficiency [33]. Symptoms such as fatigue and difficulty in accomplishing daily activities can be observed. For the diagnosis is recommended to measure the hemoglobin concentration, as well as the serum ferritin and/or transferrin receptor levels. Furthermore, there are three types of anemia: microcytic, normocytic and macrocytic, which indicate various causes [32].

When anemia is present, one of the observed problems is the reduced activity of the immune system, especially a significant reduction in the number of T cells produced and available for use. Thus, both innate and adaptative immune systems are weakened, exposing the organism to bacteria, viruses, and other pathogens [30,38]. The immune system is composed of various tissues whose primary function is protecting the organism from infections caused by several sorts of pathogens. It is divided into innate and adaptative immune systems. The innate immune system corresponds to that responsible for initiating a fast response when a pathogen enters, being acquired since birth. In turn, adaptative immunity is responsible for eliminating the pathogen during the final infection phase, being acquired late during body development [38].

Among the measures to treat anemia, adoption of a balanced diet is needed, concerning macro and micronutrients, especially a diet that includes iron, vitamin B12, folic acid, and vitamin A [32]. It is recommended to consume iron from animal sources such as red meat and fish, which have higher bioavailable iron contents when compared to plant sources [32]. Other alternatives include nutritional education practices and regular consumption of foods fortified with iron and other micronutrients [32,36].

#### 2.1.4. Zinc

Zinc is an essential mineral for the body. It is considered an oligoelement, of which approximately 90% is located in the bones and skeleton muscles, being absorbed in the small intestine through a mechanism mediated by transporters. However, recent studies suggest differences in absorption rates for different population groups according to the type of diet and the molar proportion of phytate:zinc [40]. Zinc absorption depends on its concentration in the gastrointestinal tract, i.e., it increases with its level in the diet. In addition, individuals deprived of zinc in the diet tend to show higher absorption, while individuals under a zinc-rich diet demonstrate reduced absorption [41].

During digestion, zinc is liberated from foods in the form of free ions, binds itself to endogenous secretions, and is transported to the enterocytes in the duodenum and jejunum. Some specific transport proteins can facilitate the passage of zinc through the cell membrane to the portal circulation [42]. It also can be absorbed by passive paracellular pathways, where the portal system transports absorbed zinc directly to the liver, then liberates it into the systemic circulation for delivery to other tissues [42].

Thus, zinc loss occurs mainly through the gastrointestinal tract and can also occur through urine and the body surface (scaling skin, hair, sweat). To assure better bioavailability, concomitant consumption with phytic acid should be avoided (hexa and pentaphosphate of inositol), which is the primary food factor known for impeding zinc binding to transporter cells. To counteract this, proteins of animal origin collaborate and increase the bioavailability of zinc, so favoring a better absorption [43]. However, long-term and excessive intake of zinc can cause copper deficiency and subsequent neutropenia [44].

#### 2.1.5. Calcium

Calcium (Ca) is the mineral responsible for the growth and maintenance of bone tissue and is dependent on a variety of genetic and environmental factors. It is known that heredity contributes to about 60% to 70% of the phenotypical expression of bone mineral density (BMD), one of the most important markers of bone health. In addition, environmental factors, such as diet and lifestyle, contribute to 30% to 40% of BMD. The possibility to alter these environmental factors, with consequently improved bone tissue, becomes essential in characterizing their effect on the skeleton, and for the most relevant to bone health is the dietetic Ca content [45].

Intestinal absorption of Ca is divided into two parts: active, which is mediated by vitamin D and involves the Ca-bonding protein (Ca-Bp), and passive, which can correspond to direct or facilitated diffusion (carrier-mediated) [46]. Most Ca absorption occurs in the small intestine, and this depends on the absorbing capacity, length of the intestinal segment, transit time, bioavailability, and intraluminal Ca concentration [45]. The duodenum has the highest absorptive capacity per unit length; however, most Ca is absorbed in the jejunum because of its highest total length [45].

Active intestinal absorption of Ca is primarily regulated by 1,25-dihydroxyvitamin D [1,25(OH)_2_D]. Other hormones can also influence this absorption, increasing it (parathormone PTH, growth hormone GH) or diminishing it (glucocorticoids, excess of thyroid hormones and possibly calcitonin), via interaction with the renal conversion of 25-hydroxyvitamin D (25-OHD) to 1,25(OH)_2_D, with the effect of 1,25(OH)_2_D in the intestine, or also by direct hormonal action [47]. The intestinal epithelial cells are able to closely monitor ionized calcium levels at both entry and exit points by using CaSR, which in turn activates secondary negative regulators, e.g., FGF-23, to slow down calcium absorption, thus preventing excessive uptake into the body [48].

#### 2.1.6. Magnesium

Magnesium (Mg) is the fourth cation in the human body and the second most prevalent intracellularly. It is distributed approximately 53% in bones, 27% in muscles, 19% in non-muscular soft tissues, and only 1% in the extracellular liquid; most of it in the intracellular space is bonded to various chelators, such as adenosine triphosphate (ATP), adenosine diphosphate, protein, RNA, DNA and citrate [49].

Its absorption is influenced by vitamin D and the parathyroid hormone (PTH). Administration of calcitriol to uremic patients led to normalization in Mg absorption through the jejunum; in patients with vitamin D deficiency and reposition, this showed reduction [50]. It is a nutrient present in practically all food sources; its deficiency is rarely found in normal conditions and is usually related to the presence of a subjacent disease [49].

Intracellular Mg plays an essential role in storage, transference, and utilization of energy due to the formation of Mg-ATP, which is a substrate involved in a broad variety of enzymes (e.g., phosphatases and phosphokinases) and is located in the plasmatic membrane and intracellular compartments. The activation of ATPases by specific ions results in the hydrolysis of ATP and, therefore, final control of the intracellular electrolyte content [51].

It also exerts indirect functions on protein synthesis through its action on nucleic acid polymerization, its role in the binding of ribosomes for RNA, and the synthesis and degradation of DNA. It is also involved in the anaerobic phosphorylation of glucose and the mitochondrial oxidative metabolism [52].

Either hypomagnesemic or normomagnesemic individuals are prone to several pathological conditions and require medication, mainly in those hospitalized under intensive therapy. Hypomagnesemia is more frequent in patients with normal serum creatinine and has been associated with an increased mortality rate [53].

Magnesium deficiency is characterized by severe reduction in cognitive capacity and processing, and in particular reduced attention, together with increased aggression, fatigue, and lack of concentration [52]. Other common symptoms include easy irritation, nervousness, and changing humor [53].

Magnesium helps to generate ATP and energy, eliminate ammonia in the brain related to lack of attention, and convert essential fatty acids in DHA (docosahexaenoic acid), which is related to proper functioning and structure of the brain cells. It also has an antioxidant effect, where it can reduce oxidative stress related to the physiopathology of Attention Deficit Hyperactivity Disorder (ADHD). Furthermore, magnesium can improve the sleep disorders observed with ADHD, which can adversely affect attention [54].

### 2.2. The Influence of Minerals on Pregnancy and Fetal Development

#### 2.2.1. Iodine

During pregnancy, nutrient requirements are increased to meet maternal and fetal needs. In relation to iodine, it is necessary to maintain maternal thyroid hormones at normal levels, as well as to supply them to the fetus, especially during the first trimester. In addition, this increase in requirements is essential to supply iodine clearance by the kidneys, especially during the first trimester of pregnancy [25,55].

The recommendation for iodine intake for babies (0 to 6 months) is 40 µg/day, for infants (6 to 12 months) is 50 µg/day, for children (1 to 10 years) is 60–100 µg/day, and for adolescents and adults is 150 µg/day [16]. In pregnant women, those concentrations increase to 220 µg/day, due to the extra demands of the fetus and the higher needs of thyroid hormones for both [56,57].

Iodine deficiency becomes more dangerous in women of reproductive age, pregnant and breast-feeding women, and children below three years of age, since it can cause irreversible damages. This mineral is necessary for neuron migration and myelination in the brain, and at insufficient levels it will lead to hypothyroxinemia, increasing the predisposition to irreversible damage [55].

Transport of iodine via the placenta occurs similarly to that of the thyroid gland due to mediation by the sodium iodide symporter (NIS) and pendrin (PEN), which are expressed throughout pregnancy [20]. The NIS is a transporter found during the 8th to 10th weeks of pregnancy, is located in the apical membrane of syncytiotrophoblasts, and is responsible for mediating the influx of iodine to the cells. PEN is the transporter responsible for iodine efflux and is located in the basal membrane of syncytiotrophoblasts [20].

An Australian study observed that maternal ingestion of iodine below the recommended limit (220 μg/day) was associated with alterations in the neuronal development of children, via the Bayley-III scale that analyzes cognitive, language, and motor parameters [19]. In the fetus, deficiency of this mineral can cause irreversible damage to the central nervous system, instilling a permanent condition of mental retardation since the thyroid hormones are responsible for the correct development of this process [17,58]. Iodine deficiency in the fetal phase is the principal cause of inevitable mental delay, resulting in a drop of up to 20 points in IQ (Intelligence Quotient) [2].

On the other hand, excessive iodine in the fetus induces hypothyroidism due to inhibition in the synthesis and liberation of thyroid hormones; it also affects the growth and development of the fetus, as mentioned before. These effects can be related to the capacity of iodine to interact with lipids, proteins, and nucleic acids and, thus, affect different cells in various processes, especially causing cell death by oxidative stress [17,19,20,21,56].

#### 2.2.2. Selenium

During pregnancy, adequate ingestion of selenium is fundamental since its deficiency can cause complications that affect the mother and the fetus. Among the related problems are pre-eclampsia, intolerance of glucose, alterations in the lipidic profile, mental and psychomotor delay, and other disorders [29,59,60,61].

One study reported that selenium concentrations during pregnancy were associated with its ingestion via food, then expectant mothers in the first trimester presented a higher selenium content when they had increased consumption of seafood, eggs, and bread, which are rich in this mineral [59]. Another study found that pregnant women (26 weeks) had high concentrations of selenium in their urine (55% more than estimated in the literature); thus, this mineral can have varied concentrations during pregnancy. Authors also indicated that the low consumption of this mineral can cause oxidative stress in the expectant mother and the fetus [61].

Selenium removes free radicals of oxygen, produced especially during the production of thyroid hormones [23]. The antioxidant effect of this mineral was observed in female BALB/C mice receiving high dosages of iodine. Toxicity tests showed that supplementation of 0.3 mg/L of selenium for 30 days had a protective effect on autoimmune diseases of the thyroid [21]. Hyperthyroidism was also reported in the offspring of female C57BL/6 mice, whose mothers received selenium-deficient diets, and its development was attributed to insulin resistance [24].

Another study revealed when selenium concentrations during the first trimester were 86 µg/L, the newborn presented a reduction in the development of mental and psychomotor problems [59]. Selenium deficiency in the mother also reflected intolerance to glucose and insulin resistance in her offspring [24].

In light of this, selenium also takes part in the metabolism of thyroid hormones protecting against oxidative damages, and its deficiency is associated with premature birth, miscarriage, and pre-eclampsia. One study observed that as the weeks of pregnancy progressed, the selenium concentration in the urine was reduced. Such a fact was considered because if the mother does ingest the ideal amount or is supplemented, in pregnancy as well as in breastfeeding, there is no guarantee that adequate transference via the placenta and mammary glands will occur [62].

On the other hand, another study found that when the selenium concentrations in the placenta were high, a positive association occurred with the development of problems in the neural tube of the newborn. Those authors claim that this can occur since the placenta accumulates high quantities of this mineral, so the fetus does not receive the needed amounts [63]. Selenium concentrations in the fetus are variable due to the different forms of transference via the umbilical cord. Selenoprotein P (SeP) and selenoalbumin (SeAlb) concentrations, for example, present different mechanisms of transference from the mother to the fetus and are also in different concentrations. In the mother, SeP and SeAlb concentrations are 65% and 15%, respectively, while in the umbilical cord it is around 28% for selenoalbumin. Such differences can be attributed to changes referring to the available transporters in each part [64].

Some studies show that evidence between the selenium status and adverse results on pregnancy is not yet thoroughly supported; however, this mineral seems to have a protective effect on the number of such complications since selenoproteins reduce oxidative stress and inflammation of the endoplasmic reticulum, protect the endothelium, regulate the vascular tonus, and diminish infections. Consequently, a higher selenium level would lead to a lower risk of miscarriage and premature birth, and some trials also showed that supplementation could reduce the risk of pre-eclampsia and thyroid disease after birth [65].

In another cohort study consisting of 410 mother–child pairs, assessing the status of pre-natal selenium in each trimester, during childbirth and in umbilical cord blood, positive effects were shown on the psychomotor abilities of children in the first two years of life, mainly due to the status of the mineral in the first trimester. The study used Bayley scales of child and infant development to assess psychomotor development [66].

#### 2.2.3. Iron

During pregnancy, iron deficiency is frequent because of the increased requirements, as well as of other nutrients. Thus, adequate care is necessary for its consumption, and in some cases, supplementation is recommended [34,35,36]. The iron requirements during pregnancy gradually increase over the three trimesters: during the first trimester 6.0 mg/day is recommended, in the second 19 mg/day, and in the third 22 mg/day [67]. Initially, iron is required in lower quantities and utilized for the placenta and fetus growth. After the first trimester, the needs increase due to the weight gain of the mother and the fetus, especially for the growth of their new tissues [68].

Iron deprivation in the beginning of pregnancy (first and second trimester) can lead to premature birth or low birth weight and jeopardize the health of the newborn [32,34,69]. Gestational anemia is related to a higher maternal mortality rate as well as interferes with the weight and health of the newborn [70]. It also can lead to miscarriage during the first trimester [68].

Iron demands are increased during pregnancy due to the high hemoglobin levels required for both the mother and fetus. However, when ingestion is not adequate, it is possible to observe a reduction in iron concentration as the blood volume increases [57]. Thus, identifying the status of iron in the mother at the beginning of pregnancy becomes fundamental [67]. Several studies report that reductions in fetal growth and weight are typical in newborns of iron-deficient mothers during pregnancy, and such a reduction is persistent over the course of childhood [31,34]. One study found that expectant mothers with low ferritin content (<15 µg/L) delivered iron-deficient babies [71].

Besides restricting growth and development, other health hazards arise related to iron deficiency during pregnancy such as neuronal changes and problems in myelinization, neuronal transmission, and development of the frontal cortex and basal ganglia [34,35,72,73]. Other issues observed in situations of iron deficiency during pregnancy are the increase in blood pressure of the offspring and alterations in the liver [74]. These authors observed that the animals presented a reduction in the hepatic content of iron when it was not acquired adequately from food, suggesting that this would be a compensatory mechanism to maintain body homeostasis.

Furthermore, studies have correlated the possible influence of obesity on iron levels [73]. In a study with obese pregnant women, a negative association with iron content was observed and affected important markers [72]. Gestational obesity is also associated with gestational diabetes, pre-eclampsia, hypertension, depression, induction of cesarian, and risk of congenital abnormalities [73].

In this scenario, supplementation is recommended in cases of iron deficiency in the mother, especially during the first trimester. In a cross-sectional study it was found that out of 207 evaluated pregnant women, 65.2% were supplemented during pregnancy, of which 84.4% presented iron deficiency [75]. Despite the evidence, the indication of iron supplementation during pregnancy is not a consensus in the literature due to inconsistent results [35].

In general, it is observed that iron supplementation during pregnancy is indicated in expectant mothers within the requirements or those with deficiency [38]. In a study on pregnant women over 28 weeks, it was observed that iron and folic acid (60 mg and 0.25 mg, respectively) were responsible for the increase in hemoglobin levels, besides a related reduction in underweight birth [70]. Thus, it is recommended that supplementation should always be under medical supervision since excess iron can cause immunity problems due supersaturation [30,69].

Iron overload before or during pregnancy can cause adverse effects on cellular function, especially inhibition of lymphocyte proliferation, thus requiring an individualized evaluation for supplementation [38,69]. Excess iron can be responsible for the development of oxidative stress, elevated blood viscosity, and even risk of pre-eclampsia [69]. Other authors also report that high iron concentrations can cause an increase in the incidence of insulin resistance and diabetes mellitus type 2 during pregnancy [57,76].

#### 2.2.4. Zinc

Approximately 18% [77] of pregnant women have zinc deficiency. This causes adverse effects during pregnancy and can lead to premature birth as well as in stunted growth and development. Zinc deficiency is also associated with increased risk of infections and nanism in children [78]. Research in four African countries detected the prevalence of low concentrations of plasmatic zinc varying from 45% to 83% in newborns and 52% to 82% in women of reproductive age [79]. Similar rates were found in Bangladesh, Vietnam, and Colombia [80]. Even though in lower intensity, zinc deficiency also was prevalent in Pakistan, the Republic of Maldives, Philippines, and Mexico [81].

The recommended dose of zinc for pregnant women is 11–13 mg/day. Zinc deficiencies in pregnancy are related to complications at childbirth, such as severe undernourishment, and are also linked to the regulation of T-helper cytokines, resulting in recruitment of T-helper cells that favor clinical infections and harm neonatal development, with particular actions on the immune system and brain development [82].

Zinc is one of the metallic ions most prevalent in the brain. It participates in neurogenesis, i.e., neuronal migration and differentiation, thus impacting cognitive development and helping to keep the brain functionally fit [83]. Interruptions to the immune system, such as cytokine dysregulation, and to zinc homeostasis can affect synaptic transmissions, which has been identified in individuals with autism [84]. Additionally, zinc deficiency can cause cerebral white matter injury (WMI), the primary matter of the premature brain, and is present in more than 50% of premature and extremely premature babies [85]. Its morphologic consequences are subjacent to most observed subsequent neurological deficits since the target in the cellular central area of WML is the pre-oligodendrocyte (pre-OL), the progenitor of OLs, which produce mature myelin. Lesions in the pre-OL in the premature brain leads to its failure and subsequent hypomyelination [85].

Sociodemographic variables, lifestyle, and diet also contribute to exposure of pregnant women to oligo-elements [6]. The consumption of alcohol can affect the absorption of copper, manganese, and zinc, and the latter, when detected in the urine, can potentialize teratogenic effects [6]. In a study that evaluated concentrations of minerals in the blood, umbilical cord, and urine of pregnant women, copper and zinc showed much lower levels in the cord than in the blood, suggesting that transplacental passage is limited. Other effects included diminished levels from the first to the third trimester due to hemodilution, reduction of the levels of zinc-binding protein, and hormonal alterations [62].

#### 2.2.5. Calcium

During pregnancy, the recommendation for calcium is 1000–1300 mg/day, and changes occur in the Ca metabolism that favor its transference to the fetus, including alteration in the regulating hormones of this element [47]. Thus, the intestinal absorption rate is increased, mainly from the second trimester onwards, by 27% (in non-pregnant women) to 54% in the fifth or sixth month of pregnancy, reaching 42% at term (40th gestational week) [47]. This also favors the increase in renal reabsorption and the turnover of bone Ca, as well as urinary Ca, probably due to the increased glomerular filtration rate [47]. During pregnancy, increased calcium absorption occurs and is stimulated by some hormones (vitamin D, estrogen, lactogen, and prolactin) and shows higher retention in the kidney tubules [1].

Ca is essential for many biological processes during pregnancy, including supply to the formation of bones and teeth, signal transduction, muscular contraction, enzymatic regulation, and blood coagulation [86].

During pregnancy, severe Ca deficiency can threaten the mother, increasing the risk of pre-eclampsia, which a significant cause of maternal mortality in developing countries, and it can threaten the fetus, as it can contribute to spontaneous prematurity. Pre-eclampsia can contribute to miscarriage or premature birth induced by drugs, which increases the risk of neonatal mortality [86]. Regarding premature birth, Ca deficiency can lead to restriction of intra-uterine growth, contribute to low weight at birth, and is implied in a variety of physical and cognitive consequences in the long term, including increased atrophy [87]. Pre-natal Ca ingestion can reduce the risk of premature birth and can be associated with its function in supporting fetal growth and maturity [88].

In all children, Ca serum concentrations immediately after birth drop significantly. This response is exaggerated in the premature compared with the full-term baby and is frequently referred to as early hypocalcemia of prematurity (EHP). The degree of hypocalcemia increases earlier the gestational age, with less mature babies having the lowest levels of serum calcium. However, hypocalcemia is typically asymptomatic, although several side effects have been clearly documented. It is difficult to determine the morbimortality associated with early hypocalcemia [89]. Many other complicating variables affect babies with low serum calcium concentrations, such as respiratory distress syndrome, intraventricular hemorrhage, convulsions, hypotension, metabolic acidosis, necrosing enterocolitis, and sepsis [89].

Calcium deficiency during pregnancy is typical in developing countries, populations with low ingestion, or with higher risk of gestational hypertension, and the WHO recommends Ca supplementation with 1.5–2.0 g / day from the 20th week onwards [13]. Attention should be paid to the interaction of minerals, calcium and zinc for example, in reducing the absorption of iron [90].

#### 2.2.6. Magnesium

During pregnancy, 360–400 mg/day of magnesium is recommended, and its absence can trigger severe pre-eclampsia. Use of magnesium sulphate as a prophylactic anticonvulsant in severe pre-eclampsia and its treatment have been established and demonstrably improve maternal health, and it has also proven to be a protecting agent in premature birth [91]. Despite the documented advantages to the newborn resulting from magnesium administration in early preterm pregnancies, it is still unclear if such an advantage exists for late premature birth and full-term birth [92].

Magnesium deficiency has been adversely implicated in maternal and perinatal settings, as it has been associated with risks, such as hypertensive pregnancy syndromes, leg cramps, and premature birth. Mg deficiency has also been associated with the highest Apgar scores among newborns and reduced occurrence of hypoxic-ischemic encephalopathy; however, as only 2 of the 10 randomized clinical trials were considered of highest quality, the authors concluded that there was insufficient evidence to support oral Mg supplementation during pregnancy as beneficial for the mother and/or the fetus [93].

### 2.3. The Influence of Minerals on Child Growth and Development

#### 2.3.1. Iodine

Iodine is an essential mineral for growth and development, especially during the gestational phase when its deficiency can lead to cretinism in children [19].

Iodine deficiency during the first years of a child’s life causes changes in brain development that can lead to reduced mental activity [94]. A cross-sectional study in the Netherlands found that iodine intake was 84% below that recommended for women aged 7 to 69 years, and for men it was 69% below that recommended in the same age group. In the case of children, the authors found that low iodine consumption was directly related to low parental education and low consumption of breast milk throughout the first years of life [94].

Another important point to be mentioned is that the iodine content in breast milk is higher in the postpartum period when compared with the lactation period [95]. Breast milk during the first 6 months of a newborn’s life is the only source of iodine available. One study found that the concentration of iodine present in the colostrum could predict the child’s motor development [96]. In addition, the iodine content in breast milk has been linked to geographical location. Thus, in places that offer an adequate supply of iodine to the population, mothers will be less predisposed to having a low content in their milk [95].

In childhood, iodine deficiency showed to be a risk factor for delayed development [97]. There are studies that also associate the deficiency of this mineral with autism and attention deficit and hyperactivity disorder in children [98]. A systematic review on supplementation and mental development among children suggests that light to moderate iodine deficiency at infancy has adverse effects on cognitive and motor performance, showing a drop of 6.9 to 10.2 IQ points [55].

In another cohort study performed over 15 years, sons of mothers with light iodine deficiency showed reduced results of alphabetization (orthography, grammar, and reading) at 9 years of age, and this persisted throughout adolescence. Brain development is not concluded during the intrauterine period, and adequate ingestion of iodine during infancy is necessary for the continuous and ideal development of this organ [99]. It is important to emphasize that thyroid hormones are involved in growth and development of the brain and central nervous system from the 15th week of pregnancy onwards until 3 years of age [100].

Iodine has also shown to play a fundamental role in diabetes mellitus type 1 (DM1). A study observed that newborns receiving iodine immediately after birth until the sixth day of life presented a reduced incidence of DM1. This was attributed to the capacity of iodine to reduce the content of pancreatic insulin and insulin secretion induced by glucose; however, little is known about all mechanisms involved in this process [15].

#### 2.3.2. Selenium

Selenium acquired during pregnancy is maintained in reserves in the newborn, especially those born at an adequate gestation time, since the buildup of this mineral in the uterus occurs in the third trimester. However, such reserves are low, and it is necessary to acquire selenium via maternal milk, which also has a variable content according to the quantity acquired by the mother via diet [16,43,101].

Selenium occurs in higher quantities in the colostrum compared with the subsequent milk. However, generally, the concentration of this mineral in the maternal milk is sufficient to fulfill the baby’s requirements (either born at full-term or pre-term). After the exclusive breastfeeding period, it is necessary to introduce food sources of this mineral [16,43,101].

When in low concentrations in the newborn, especially premature, they are at higher risk for developing diseases due to increased susceptibility to oxidative stress. Among the common diseases are retinopathy, bronchopulmonary dysplasia, and other lung disorders. Besides maternal milk, supplementation with milk formulas is an alternative way to supply the needs of this mineral, depending on the available content in the formula [43].

On the other hand, excess selenium has also demonstrated adverse effects on health, and it can trigger cardiovascular problems, dyslipidemias, and insulin resistance [29]. More visible problems can occur such as alterations in the keratin layer of fingernails and hair, skin problems, and gastrointestinal disorders [16]. A study showed that animals receiving supplementation of selenium (0.5 ppm) for three weeks during pregnancy and three weeks during breastfeeding presented offspring with pancreatic dysfunction. These authors claim that this mineral has a central role in the pancreatic endocrine function, with the regulation of the β-pancreatic cells in the redox status [102].

Excess selenium is known as selenosis and is characterized when its values are above 400 µg/day (tolerable daily level) in adults. Among the characteristic symptoms of the disease are episodes of vomiting, diarrhea, neuronal disorders, hair loss, and infertility [22,23,76,103]. Cases of selenium toxicity occur in lower proportions and in isolated regions where its presence in the soil is abundant [23]. Increased consumption of selenium also causes changes in the secretion (increase) of insulin and promotes greater predisposition to the development of obesity and inflammation [104].

#### 2.3.3. Iron

A study observed that iron deficiency during pregnancy was responsible for the development of neuronal alterations in newborns, which persisted until adulthood despite supplementation [105]. Such neuronal problems occur mainly in the initial phase of life since the brain needs a high iron content to develop, similar to the liver. Iron plays an essential role in synaptogenesis, synthesis of neurotransmitters (e.g., dopamine, serotonin, and noradrenalin), synthesis of neurotrophin, and in myelinization, as cited above [33,71]. Thus, the lower its available content, the more severe the damage, with the worst being irreversible [16].

Besides deficiency during pregnancy, reduced iron reserves also occur at birth, and maternal milk can be poor in this mineral [11]. The incidence of iron deficiency in babies is 73%, and its evolution to anemia is very high. If anemia occurs in the first year of life, studies demonstrate that delayed speech is possible. Since iron is essential for myelinization, transmission in these sensor systems is affected by the lack of this mineral, and this can lead to delayed development, reduced school performance, behavioral disorders, ADHD, and risk of cerebrovascular accident (CVA) in healthy young children [90].

Even in cases of iron supplementation, brain alterations can be irreversible, especially when iron deficiency is present during phases of neurogenesis and cellular differentiation of the brain areas [32]. Studies also show that in situations of iron deficiency (without gestational anemia), damages can occur to the cellular processes of white cells and in regions of gray matter of the central nervous system, which persist in adult life [31,106].

Such damages can be attributed to altered synthesis of neurotrophin, a growth factor dependent on iron, which is responsible for protecting the neurons involved in learning processes and behavioral development [71]. The effects on memory were observed in a study on adult animals descendent from mothers with iron deficiency during pregnancy. These authors attribute such damages to the presence of cytochrome oxidase, a marker that has neuronal activity in the ventral hippocampus [34].

This region of the hippocampus develops faster during the last trimester of pregnancy and is the area responsible for memory storage (intellectual and spatial). Its growth occurs until two years of age; therefore, this is a crucial phase for adequate ingestion of iron [33]. Other authors also observed that iron deficiency during pregnancy resulted in changes to the brain of the new-born, and this was attributed to the alterations that occur in the composition of lipids that are indispensable to the fluidness of the cell membrane, as well as in its functionality [31].

Newborns can also receive iron supplementation, but this should be followed on an individual basis; thus, dosages and duration of administration will vary according to the birth weight [106]. Supplementation occurs in specific situations, as the newborn is able to meet iron needs by recycling hemoglobin since breast milk offers a lower iron content [107].

Another important point is that iron, when in excess, can impair the absorption of other minerals such as copper and zinc, as it acts directly on their transporters, so it is important to check iron concentrations when it is supplemented [107]. For newborns, exclusive breastfeeding is recommended until 6 months [32].

Anemia has been associated with poor motor development and with irreversible cognitive defects [97]. A greater concern arises because there is evidence that iron-deficient and obese children can develop neurodegenerative diseases due to elevation of hepcidin, a protein regulator of iron, that is required for the adequate functioning of neurons, and it is necessary for insulin and its receptor signal in the brain system [90].

#### 2.3.4. Zinc

Other clinical manifestations related to zinc deficiency in the first days of life are reduced natural barrier functions of the skin and mucosas. It also has deleterious effects on the immune system, favoring pathologies related to the skin, lungs, and gastrointestinal tract that can persist until adult life [108]. Thus, zinc deficiency is also associated with negative outcomes such as increased morbimortality at birth, increased severity of infectious diseases, stunted growth, and physiological alterations (anorexia, hypogonadism, hypogeusia (reduced sense of taste), dermatitis, dysfunctions of the immune system, and oxidative and neuropsychological damages) [109]. Zinc deficiency is also associated with diarrheic manifestations and predisposition to fever convulsions in children between 6 days and 6 months of age [110].

Studies on experimental animals show that gestational zinc deficiency jeopardized the learning ability and reduced the attention and memory in the offspring. Data in humans are still inconclusive; however, data reveal that this mineral is relevant to the brain function of babies and older children since the hippocampus has high zinc concentration and is also responsible for memory [90]. The human immune system is very sensitive to zinc deficiency. The recommendation for breastfeeding children is 3 mg/day, as subclinical zinc deficiency jeopardizes the formation of cell mediators of innate immunity, such as phagocytosis by macrophages and neutrophils and interference with the activity of NK cells; thus, adequate levels can reduce inflammatory processes and susceptibility to infection [42].

Zinc deficiency is considered a public health problem since nutritional scarcity in the first years of life involves multiple etiologies. Deficiency is related to the decline of zinc concentrations in maternal milk, after the first 6 months of breastfeeding, or is associated with the low ingestion of this mineral in complementary food, in the case of breastfeeding; increased physiological requirements of zinc during pregnancy, in the lactation and growth phase (infancy and adolescence); adoption of diets poor in proteins of animal origin and rich in phytates and/or with high energy value; reduced consumption of food caused by reduced mobility, which contributes to diminishing energy needs, dental problems and swallowing difficulty in the elderly, as well as deficiency of other nutrients such as vitamin A and iron [111,112].

#### 2.3.5. Calcium

After birth, Ca absorption and its urinary excretion return to pre-pregnancy levels. Some studies report that during breastfeeding a reduction in urinary excretion of Ca and phosphate occurs [113]. One cohort study suggests that maternal ingestion of calcium can reduce risks of emotional problems and hyperactivity in children 5 years of age [114].

Babies born with calcium deficiency have a higher risk of nutritional rickets, even in the absence of hypovitaminosis D, presenting pain, deformity, and fractures throughout life that may persist through adolescence and adulthood. In the adult stage of life, rickets is replaced by osteomalacia, and experimental evidence implies that vitamin D supplementation is probably necessary, even if there is no vitamin D deficiency, in children, 210–500 mg/day of calcium is recommended for physiological maintenance [115].

Metabolic Bone Disease (MBD) is a disorder commonly observed in newborn with very low weight (weight at birth <1.500 g), with a higher incidence in those with extremely low weight (<1.000 g). MBD is characterized by biochemical and radiological findings related to bone demineralization. Various prenatal and post-natal risk factors have been associated with MBD in prematurity, although the primary pathogenic mechanism is represented by reduced placental transfer of calcium and phosphate [116,117].

Like other body tissues, oral structures are also affected by premature birth. Enormous emotional, physical, and financial tolls are put on families, together with medical systems. Although the pathogenesis of dental defects remains uncertain, it is probable that systemic disorders and local factors contribute to the etiology. Calcium metabolism disorder is considered as a possible systemic factor in the pathogenesis of dental defects since teeth are formed by mineralization of the protein matrix around 4 months of pregnancy and are not complete until the end of adolescence, and most reserves of calcium and phosphorous are accumulated in the third trimester of pregnancy. Therefore, a very low weight baby will not have accumulated such stocks. Thus, the enamel in prematurely born children is immature at birth, which leads to enamel hypoplasia resulting from development anomalies [118].

Adjusting for body size, more calcium is recommended in the diets of newborns, babies, and children to support growth, aged people to improve osteoporosis, and pregnant and lactating women to fulfill the increased physiological requirements [119].

#### 2.3.6. Magnesium

There are few studies on the prevalence of magnesium deficiency in infancy and first infancy. It is known that Mg deficiency can lead to hypoparathyroidism, hypocalcemia, and impaired bone growth in children, and this can be more severe under two years of age, during the fastest growth and development phase. The mechanism involved is associated with this mineral taking part in the parathyroid gland and bone remodeling [13].

The effectiveness and safety of pre-natal Mg supplementation in the prevention of brain paralysis has been documented. The recommended dose is 30–80 mg/day, which is considered effective without hindering safety aspects. This dose does not increase neonatal mortality and other neonatal suspect complications, such as neonatal choking, spontaneous intestinal perforation, necrosing enterocolitis, and food intolerance. Thus, Mg supplementation is considered easily accessible, cheap, and is proposed as an obligatory part of handling premature births [120].

Table 1 summarizes the information regarding the general characteristics, influence of minerals on pregnancy and fetal development, and growth and development described in the text.

## 3. Conclusions

Based on our review, we observed that, mainly in the second and third trimesters of pregnancy, there is a reserve of nutrients in the fetus that can be used after birth. In this phase, until the child is 3 years of age, there is growth and development of the brain and central nervous system, which is why there are many problems associated with brain function under nutritional deficit, such as hyperactivity, attention deficit, autism, speech delay, and memory problems.

For a child to grow and develop properly, it is necessary to start planning pregnancy from conception. Many doubts still remain about how some of these processes occur, since there are variations not only in dietary deficiencies, but also in interactions between nutrients and environmental, sociocultural, and sociodemographic factors, in addition to varied concentrations depending on the compartment (e.g., blood, placenta, umbilical cord, and urine).

In this way, further research is necessary, mainly with regard to the transfer of nutrients between the mother and the baby, considering all these factors presented.

## Figures and Tables

**Table 1 molecules-25-05630-t001:** General characteristics, influence of minerals on pregnancy and fetal development, and influence of minerals on child growth and development.

Mineral	General Characteristics	Influence of Minerals on Pregnancy and Fetal Development	Influence of Minerals on Child Growth and Development
Iodine	thyroid hormones [15,16] brain development [55] redox balance [20] relationship with selenium [21]	neuronal development of children [19] mental retardation [17,58] fetal hypothyroidism [17,19,20,21,56]	delay in development [97] association with autism [98] association with attention deficit [98] association with hyperactivity disorder [98] drop in IQ points [2,55] results of alphabetization [99]
Selenium	antioxidant activity [21] action in the immune system [16,26] present in the thyroid gland [22]	intolerance to glucose [29,59,60,61] alterations in the lipidic profile [29,59,60,61] mental and psychomotor delay [29,59,60,61] oxidative stress in the mother and the fetus [61] premature birth, miscarriage [62] problems in the neural tube of the newborn [63]	oxidative stress in the premature [43] triggers cardiovascular problems [29] insulin resistance [29] neuronal disorders [22,23,76,103] secretion (increase) of insulin [104]
Iron	oxygen transport [30,31,32] production of erythrocytes [30,31,32] transport of enzymes [30,31,32] maintenance of the immune system [30,31,32] synaptogenesis [33,71] synthesis of neurotransmitters [33,71] synthesis of neurotrophin [33,71] myelinization [33,71]	premature birth or low birth weight [32,34,69] miscarriages during the first trimester 68] reduction in fetal growth and weight [31,34] neuronal changes [34,35,72,73] association is negative in obese pregnant [72] oxidative stress, risk of pre-eclampsia [69] incidence of insulin resistance in pregnancy [57,76] diabetes mellitus type 2 in pregnancy [57,76]	delayed speech [90] delays in development [90] behavioral disorders [90] obese children, neurodegenerative diseases [90]
Zinc	regulation of T-helper cytokines [82] participates in neurogenesis [83] cognitive development [83] maintains brain function [83] located in the bones and skeleton muscles [40]	premature birth [78] complications at childbirth [82] neurological deficits [85]	infections and nanism [78] functions in natural barriers in the skin and mucosas [108] morbimortality at birth [109] increased severity of infectious diseases [109] growth deficit [109] physiological alterations [109]
Calcium	functions in bone tissue [45] signal transduction [86] muscular contraction [86] enzymatic regulation [86] blood coagulation [86]	premature birth [86] pre-eclampsia [86] restriction of intra-uterine growth [87] low weight at birth [87]	nutritional rickets [115] bone demineralization [116,117] dental enamel hypoplasia [118] emotional problems and hyperactivity [114]
Magnesium	formation of Mg-ATP [51] protein synthesis [52] synthesis and degradation of DNA [52] anaerobic phosphorylation of glucose [52] mitochondrial oxidative metabolism [52] antioxidant effect [54] part of the parathyroid gland [13] bone remodeling [13]	severe pre-eclampsia [91] premature birth [93] apgar scores in the newborn [93] hypoxic-ischemic encephalopathy [93]	hypoparathyroidism [13] hypocalcemia [13] impaired bone growth [13] cognitive capacity and processing [52] lack of concentration [52]

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
