# Peer review of "Minerals in Pregnancy and Their Impact on Child Growth and Development"

_molecules, 2020, doi:10.3390/molecules25235630_

Round 1

Reviewer 1 Report

The manuscript presents interesting topic, however in my opinion is not well designed and described. I have some comments:

  1. The title suggests that all micronutrients such as minerals and vitamins are described in the manuscript, however Authors included only selected elements.
  2. Why only Se, Fe, Zn, Ca and Mg were included ? it should be explained.
  3. What kind of the article it is? The structure of the manuscript should be specified. In my opinion there is a lack of the part “methods” to establish the including and excluding criteria. The name of the subtitles (parts) should be changed and they should correspond to the structure of the article.
  4. It will be better to present the most current results and limit the cited articles to the last 5-10 years.
  5. Generally, characteristic of the each element is too long; Authors should focus on the relation between elements and pregnancy, fetal development and child growth.
  6. It will be better to divide the description on the separate parts: 1) the influence of elements on pregnancy and fetal development and 2) the influence of element on child growth and development
  7. It seems that the content partly is duplicated in point 2 (micronutrients) and point 3 (discussion).
  8. The conclusion is too general and contains known facts.

Reviewer 2 Report

This is an interesting and complete review of the behavior of several micronutrients in pregnancy and their impact on child growth and development. When the nutritional status of the patient is not satisfactory, important maternal and neonatal complications could occur; when the pattern is insufficient, that will cause deficits in neonatal development, inclusive in brain function.

In my opinion, the major concerns about the manuscript are:

1. Methodological problem: there is no description of the methodology used in the search for the most relevant literature, inclusion and exclusion criteria for the articles found. 

2. The reference is incomplete about some components in specific obstetric pathologies, such as gestational diabetes. 

Round 2

Reviewer 1 Report

Authors have corrected the manuscript according to all my comments.

I have not any other comments.